# Isolation of *Bacillus subtilis* and *Bacillus pumilus* with Anti-*Vibrio parahaemolyticus* Activity and Identification of the Anti-*Vibrio parahaemolyticus* Substance

**DOI:** 10.3390/microorganisms11071667

**Published:** 2023-06-27

**Authors:** Ning Jiang, Bin Hong, Kui Luo, Yanmei Li, Hongxin Fu, Jufang Wang

**Affiliations:** 1School of Biology and Biological Engineering, South China University of Technology, Guangzhou 510006, China; 202020148731@mail.scut.edu.cn (N.J.); 202010108485@mail.scut.edu.cn (B.H.); 15521120134@163.com (K.L.); 202010108442@mail.scut.edu.cn (Y.L.); hongxinfu@scut.edu.cn (H.F.); 2Guangdong Key Laboratory of Fermentation and Enzyme Engineering, South China University of Technology, Guangzhou 510006, China

**Keywords:** probiotics, *Bacillus*, *V. parahaemolyticus*, anti-*Vibrio parahaemolyticus* substance, aquaculture

## Abstract

The adoption of intensive farming has exacerbated disease outbreaks in aquaculture, particularly vibriosis caused by *Vibrio parahaemolyticus*. The use of probiotics to control *V. parahaemolyticus* is recognized as a good alternative to antibiotics for avoiding the development of antibiotic-resistant bacteria. In this study, two strains of B. HLJ1 and B. C1 with strong inhibitory activity on *V. parahaemolyticus* were isolated from aquaculture water and identified as *Bacillus subtilis* and *Bacillus pumilus*, respectively. Both B. HLJ1 and B. C1 lacked antibiotic resistance and virulence genes, suggesting that they are safe for use in aquaculture. In addition, these two strains can tolerate acid environments, produce spores, secrete extracellular enzymes, and co-aggregate as well as auto-aggregate with *V. parahaemolyticus*. B. HLJ1 and B. C1 produced the same anti-*V. parahaemolyticus* substance, which was identified as AI-77-F and belongs to amicoumacins. Both B. C1 and B. HLJ1 showed inhibitory activity against 11 different *V. parahaemolyticus* and could effectively control the growth of *V. parahaemolyticus* in simulated aquaculture wastewater when the concentration of B. C1 and B. HLJ1 reached 1 × 10^7^ CFU/mL. This study shows that B. HLJ1 and B. C1 have great potential as aquaculture probiotics.

## 1. Introduction

Aquaculture has emerged as one of the sectors with the greatest growth rates due to its ability to supply high-quality animal protein [1]. However, the adoption of intensive farming to meet market demand has resulted in increased frequency of disease outbreaks in aquaculture, particularly vibriosis, which causes enormous economic losses every year [2]. *Vibrio parahaemolyticus*, one of the primary causative agents of vibriosis, can cause acute hepatopancreatic necrosis disease (AHPND) [3]. The global outbreaks of AHPND have led to a huge reduction in shrimp production, resulting in a tremendous loss of national income in the affected countries, such as China, Thailand, India, America, and so on [4]. For example, the AHPND outbreak in Thailand caused a drop in production from 611,194 t in 2011 to ca. 200,000 t by 2014/15, which caused consequential national losses of over USD 7.8–11 billion [5]. Traditionally, antibiotics have been used to prevent and treat vibriosis in aquaculture, but overuse of antibiotics has led to the development of antibiotic-resistant bacteria [6]. Alternatives to antibiotics have been researched in recent decades, and probiotics are considered to be environmentally friendly and promising alternatives [7].

In aquaculture, probiotics generally refer to live, dead, or bacterial cell components that can inhibit pathogens, promote nutrient absorption and improve immunity in aquaculture animals, and also purify the water quality of aquaculture water [8]. Most reported aquaculture probiotics belong to *Lactobacillus* and *Bacillus* owing to their wide distribution and adaptability, and some can produce antibacterial compounds [9]. It was found that shrimp feed containing 4 × 10^8^ CFU/g of *Lactobacillus plantarum* SGLAB01 can reduce mortality when shrimps were infected by *V. parahaemolyticus* [10]. When *L. plantarum* ATCC 8014 was added at a concentration of 1 × 10^7^ CFU/mL, the number of *V. parahaemolyticus* and the mortality of pacific oysters dramatically decreased [11]. *B. subtilis* AQAHBS001, isolated from the intestine of pacific white shrimp in Thailand, can inhibit the growth of *V. parahaemolyticus*, and 1 × 10^7^ or 1 × 10^9^ CFU/kg in the feed can significantly improve the resistance of shrimp to *V. parahaemolyticus* [12]. The addition of 1 × 10^8^ CFU/g of *B. coagulans* ATCC 7050 in feed can enhance the growth of *Litopenaeus vannamei* and also resistance to *V. parahaemolyticus* [13]. *B. licheniformis* ATCC 11946 at 10^8^ CFU/g in feed can increase the survival rate of pacific white shrimp which was infected with *V. parahaemolyticus* [14]. Supplementing the diet of white shrimp with 1 × 10^8^ CFU/g of *B. subtilis* WB60 reduces mortality in shrimp infected with *V. parahaemolyticus* [15].

A good probiotic must be safe for aquatic organisms in aquaculture, which can directly inhibit pathogenic bacteria or indirectly improve the resistance of aquatic organisms to diseases by purifying water quality and improving the nutrient absorption and immunity of aquatic organisms [1]. So, probiotics used in aquaculture are usually tested for their inhibitory activity against pathogens, tolerance to acid and high temperatures, and safety [16,17]. In addition, the capacity to secrete extracellular enzymes, to auto-aggregate, and to co-aggregate with pathogens are also criteria to assess probiotics in aquaculture [18].

In this study, almost 200 strains were isolated from aquaculture water samples, and a total of 12 strains with inhibitory activity against *V. parahaemolyticus* were selected using the double-layer plate method. Then, selected strains were co-cultured with genetically modified *V. parahaemolyticus*, which constitutively expressed GFP as an indicator for evaluating inhibitory effect in a liquid environment. Subsequently, the potential probiotics were identified by 16S rDNA amplification and safety, and other probiotic properties were also evaluated. Furthermore, the anti-*V. parahaemolyticus* substances produced by the potential probiotics were extracted, purified, and identified. Finally, the inhibitory activity of potential probiotics against 11 different *V. parahaemolyticus* was tested, as well as the inhibitory effect on *V. parahaemolyticus* in simulated aquaculture water.

## 2. Materials and Methods

### 2.1. Sample Collection and Bacteria Isolation

Aquaculture water samples were obtained from Foshan, Guangdong Province, China. The water samples were treated at 80 °C for 20 min; water samples with high-temperature treatment and original water samples were sequentially diluted with sterile PBS in a gradient manner (10^−1^, 10^−2^ and 10^−3^). Then, 100 μL of serial dilutions were spread on Luria–Bertani (LB) agar plates and DeMan–Rogosa–Sharpe (MRS) agar plates. All agar plates were incubated at 37 °C for 24 h. The purified single colony was inoculated into 10 mL LB liquid medium and cultured at 37 °C for 16 h. The bacterial solution was mixed with 60% glycerin by volume of 2:1 in a cryopreserved tube and stored at −80 °C.

### 2.2. Screening of Anti-V. parahaemolyticus Bacteria

#### 2.2.1. Primary Screening for Strains by Double-Layer Plate Method

The *Vibrio* strain used for the primary screening was *V. parahaemolyticus* ATCC17802, which carries the *trh* virulence genes. The *trh* are the main virulence factors responsible for cardiotoxicity, enterotoxicity, and erythrocyte hemolysis [19,20]. The inhibitory activity against *V. parahaemolyticus* of the isolated strains was tested using a double-layer plate method as previously described [21], with minor modifications. First, 10 mL of LB agar was spread on the bottom of the plate, and then 10 mL of LB agar containing 10^7^ CFU/mL of *V. parahaemolyticus* was spread on top. Afterwards, 8 mm wells were punched out in the top agar layer using an Oxford cup, and 100 μL of the isolated strains were added to the wells. After incubation at 37 °C for 24 h, the diameter of the inhibition circle was measured.

#### 2.2.2. Second Screening for Strains by Co-Culture Method

To facilitate the measurement of *V. parahaemolyticus* growth in the mixed culture system, *V. parahaemolyticus* ATCC17802 harboring puC19-GFP plasmid with a GFP overexpression (named as 17802 [GFP]) was obtained by genetic engineering. The 17802 [GFP] was co-cultured with isolated strains in 96 microwell plates (final volume: 200 μL) as Goulden et al. described [22]. The 17802 [GFP] was set at a concentration of 10^4^ CFU/mL and the final concentration of isolated strains was at 10^5^, 10^6^, and 10^7^ CFU/mL, respectively.

Using the LINEST function in Excel, the relationship between fluorescence and the number (lg CFU/mL) of 17802 [GFP] was examined, and there was a multiplicative power relation between fluorescence and the 17802 [GFP] number at the first 12 h (y = 6.6895x^0.0593^; R² = 0.9835). Based on the fluorescence signal and the relationship between fluorescence and the growth of *V. parahaemolyticus*, the number of *V. parahaemolyticus* in the experimental group (N_1_) and the number of *V. parahaemolyticus* in the control (N_0_) were calculated and the inhibition rate of *V. parahaemolyticus* was calculated as follows:$$\mathrm{Inhibition}\mathrm{rate}\mathrm{of}V.parahaemolyticus=(1-\frac{\mathrm{lg}\;N_{1}-6.6895}{\mathrm{lg}\;N_{0}-6.6895})\times 100\%$$

The inhibitory effect of strains on *V. parahaemolyticus* was classified into three categories: low inhibitory activity (25% ≤ Inhibition rate of *V. parahaemolyticus* < 50%), moderate inhibitory activity (50% ≤ Inhibition rate of *V. parahaemolyticus* < 75%), and high inhibitory activity (Inhibition rate of *V. parahaemolyticus* ≥ 75%).

### 2.3. Identification of Strains

The DNA of the isolated strains was extracted according to the manufacturer’s instructions by using the Bacteria DNA Isolation Mini Kit (Vazyme, Nanjing, China). The 16S rDNA gene was amplified by a polymerase chain reaction (PCR) using primers 27F (AGAGTTTGATCMTGGCTCAG) and 1492R (GGTTACCTTGTTACGACTT). Many *Bacillus* species have been observed to display fairly conserved 16S rRNA sequences compared to other genera, and the use of this taxonomic marker is sometimes insufficient to define the species [23]. For this reason, the analysis of additional housekeeping genes, such as *gyrA*, *gyrB*, and *pycA*, has been recently incorporated in order to obtain a more reliable phylogeny [24,25,26,27]. We additionally used the *gyrB* and *pycA* genes for the identification of *Bacillus* strains. Partial sequences of *gyrB* and *pycA* were amplified using specific primers [28,29]. The pure DNA products were delivered to Sangon Biotech Co., Ltd. (Shanghai, China) for sequencing. The nucleotide sequences were subsequently compared with the available sequences in the database of the National Center for Biotechnology Information (NCBI) using the Basic Local Alignment Search Tool (BLAST) program (https://blast.ncbi.nlm.nih.gov/Blast.cgi, accessed on 15 June 2022). Multiple sequence alignments were obtained by using the CLUSTALW software included in the MEGA-X software package. The phylogenetic tree of isolated strains was constructed based on the comparison of other bacterial sequences using neighbor-joining (NJ) analysis with 1000 bootstrap replicates.

### 2.4. Safety of the Candidate Strains

#### 2.4.1. Antibiotic Susceptibility

The antibiotic susceptibility of the isolated strains was determined using the Clinical and Laboratory Standard Institute’s (CLSI) broth microdilution method. The antibiotics recommended by the European Food Safety Authority Panel on Additives and Products or Substances Used in Animal Feed (FEEDAP 2012) include teicoplanin, vancomycin, chloramphenicol, tetracycline, erythromycin, gentamycin, and kanamycin. According to the guidelines of CLSI, the isolated strains were classified as sensitive, intermediate, or resistant.

#### 2.4.2. Virulence Factor Genes

The enterotoxin genes *nheA*, *nheB*, *nheC*, *hblA*, *hblC*, *hblD*, and *entFM* were identified by PCR while double-distilled water was used as a negative control. The sequences of all primers used were designed according to the previous study [30]. The PCR results were examined using agarose gel electrophoresis (1% *w*/*v*) and gels were imaged under UV light after electrophoresis.

### 2.5. Acid Tolerance and Spore Production of the Candidate Strains

#### 2.5.1. Acid Tolerance Assay

The acid tolerance of the strains was measured as previously described [31]. The strains were incubated in LB medium at 37 °C for 2 h at pH 3–7 (pH 7 was used as a control). The survival rate was calculated by the plate count method.
$$\mathrm{Survival}\mathrm{rate}=\frac{\mathrm{Average}\;\mathrm{number}\;\mathrm{of}\;\mathrm{colonies}\;\mathrm{in}\;\mathrm{the}\;\mathrm{experimental}\;\mathrm{group}\;(\mathrm{lg}\;\mathrm{CFU}/\mathrm{mL})}{\mathrm{Average}\;\mathrm{number}\;\mathrm{of}\;\mathrm{colonies}\;\mathrm{in}\;\mathrm{the}\;\mathrm{control}\;\mathrm{group}\;(\mathrm{lg}\;\mathrm{CFU}/\mathrm{mL})}\times 100\%$$

#### 2.5.2. Spore Production Assay

After the strains were incubated at 37 °C for 24, 36, 48, 60, 72, and 96 h, the cultures were diluted at 10^−4^, 10^−5^, and 10^−6^, and then treated at 80 °C for 20 min. The number of survival bacteria in the experimental group (N_1_) and the number of bacteria in the control (N_0_) were calculated using the plate count method:$$\mathrm{Bacterial}\mathrm{survival}\mathrm{rate}=\frac{\mathrm{lg}\;N_{1}}{\mathrm{lg}\;N_{0}}\times 100\%$$

### 2.6. Extracellular Enzyme Activity and Aggregation Capacity

#### 2.6.1. Extracellular Enzyme Activity

The extracellular enzyme activity of the strains was evaluated qualitatively using starch agar, CMC-Na agar, and skim milk agar, as described by Zhou et al. [32]. The presence of a halo or clear zone indicated a positive result.

#### 2.6.2. Auto-Aggregation and Co-Aggregation Assay

Auto-aggregation and co-aggregation assays were described by Mustafa et al. [33]. The isolated strains were incubated in LB medium for 24 h at 37 °C and centrifuged at 5000 rpm for 10 min. The pellet was washed twice in phosphate-buffered saline (PBS) and then adjusted to a cell concentration of 10^8^ CFU/mL. The auto-aggregation rate was calculated as follows:$$Auto-aggregation=(1-\frac{A_{1}}{A_{0}})\times 100\%$$
A_0_ = Absorbance at 0 h (OD 600 nm)
A_1_ = Absorbance at 2 h, 24 h (OD 600 nm)

The isolated strains and *V. parahaemolyticus* were mixed for 10 s in a 15 mL conical tube, followed by incubation at 37 °C for 2 and 24 h. The co-aggregation rate was calculated as follows:$$\mathrm{Co}-\mathrm{aggregation}=\frac{1-A\mathrm{mix}}{(A\mathrm{isolate}+A\mathrm{pathogen})/2}\times 100\%$$
A_mix_ = Absorbance at 2 h, 24 h (OD 600 nm)
A_isolate_ = Absorbance of isolated bacteria at 0 h (OD 600 nm)
A_pathogen_ = Absorbance of pathogenic bacteria at 0 h (OD 600 nm)

### 2.7. Identification of Anti-V. parahaemolyticus Substance

A three-step procedure was applied to purify and identify the anti-*V. parahaemolyticus* substance, including: (i) The 2 L supernatant was extracted three times with an equal volume of ethyl acetate and then extracts were concentrated by rotary evaporation at 30 °C under reduced pressure. The above-mentioned dried ethyl acetate extract was re-dissolved in 5 mL methanol and the solvent was filtered through a 0.22 μm nylon membrane. (ii) The crude extracts were chromatographed on Sephadex LH-20 (16 mm × 70 mm), eluting with methanol at a flow rate of 2 mL/min. A total of 280 mL of eluate was collected (14 mL per tube), and the obtained eluate from each tube was tested for inhibitory activity against *V. parahaemolyticus*. The active fractions were combined and then injected into a C18 column (Agilent ZORBAX 300SB-C18, 4.6 × 150 mm, 5-Micron, 1 mL/min). The mobile phase consists of 0.1% trifluoroacetic acid (TFA)-water (solvent A) and 0.1% TFA-acetonitrile (solvent B). The gradient elution procedure was as follows: from 0 to 20.0 min with 5 to 80% solvent B, from 20.0 to 21.5 min with 80% to 100% solvent B, from 21.5 to 27.0 min with 100% solvent B, from 27.0 to 27.5 min with 100% to 5% solvent B, and from 27.5 to 30 min with 5% solvent B. The active peaks, where the anti-*V. parahaemolyticus* substances were located, were identified. (iii) The active compounds were analyzed by LC-MS (Thermo Scientific™ Accucore™ aQ C18-LC coupled with Thermo Scientific Q-Exactive ESI-MS, Waltham, MA, USA). The MS analysis uses a scan range of 50–750 (*m*/*z*) with an accuracy of 0.002% for ± mass determination.

### 2.8. Inhibitory Activity against Different V. parahaemolyticus

The inhibitory activity of two strains against 11 different *V. parahaemolyticus* was tested by the double-layer plate method. After incubation at 37 °C for 24 h, the diameter of the inhibition zone was measured.

### 2.9. Anti-V. parahaemolyticus Effect in Simulated Aquaculture Wastewater

The two strains were co-cultured with *V. parahaemolyticus* 17802 in simulated aquaculture wastewater, respectively. According to a previous report by Zhang et al. [34], synthetic wastewater media (SWM) were used to simulate aquaculture wastewater. *V. parahaemolyticus* was inoculated into 100 mL SWM at approximately 10^4^ CFU/mL, while two strains were inoculated at 10^5^, 10^6^, and 10^7^ CFU/mL, respectively. The number of *V. parahaemolyticus* was measured using thiosulfate citrate bile salts sucrose agar plates (TCBS).

## 3. Results

### 3.1. Screening of Anti-V. parahaemolyticus Bacteria

#### 3.1.1. Primary Screening by Double-Layer Plate Method

As described in the material and methods section, almost 200 strains were isolated from aquaculture water sample and inhibitory activities against *V. parahaemolyticus* were assessed. Inhibitory activity was judged by the formation of inhibition circles on the plate and the results are shown in Table 1. A total of 12 strains showed obviously inhibitory activity against *V. parahaemolyticus*. Among those strains, B. C1 showed the strongest inhibitory effect, producing an inhibitory circle of more than 20 mm. The practical application of the strains in aquaculture is in a liquid environment, so the 12 strains with *V. parahaemolyticus* inhibitory activity in Table 1 were selected for the second screening experiment and co-cultured with *V. parahaemolyticus* in liquid medium.

#### 3.1.2. Second Screening for Strains by Co-Culture Method

Both 17802 [GFP] and wild type *V. parahaemolyticus* ATCC 17802 have identical growth profiles. The isolated strains were co-cultured with 17802 [GFP] at 10^5^, 10^6^, and 10^7^ CFU/mL, respectively. When the initial concentration of isolated strains was set at 1 × 10^5^ CFU/mL, the strains had relatively low inhibitory activity against *V. parahaemolyticus* and the inhibition rates were less than 25%—except for strain JZ12 which showed low inhibitory activity against the growth of *V. parahaemolyticus* with the inhibition rate against *V. parahaemolyticus* at 40.88%. The strain JZ12 shows moderate inhibitory activity against the growth of *V. parahaemolyticus* and the inhibition rate of JZ12 against *V. parahaemolyticus* was 51.29% when the initial concentration of isolated strains increased to 1 × 10^6^ CFU/mL. In this initial concentration (1 × 10^6^ CFU/mL), the inhibition rates of two strains (B. C1, B. HLJ1) against *V. parahaemolyticus* were 32.61% and 37.88%. At the highest initial concentration (1 × 10^7^ CFU/mL), strains JZ12 and M. JZ5 showed moderate inhibitory activity against *V. parahaemolyticus* and the inhibition rates of *V. parahaemolyticus* were 61.41% and 74.67%, respectively. The strains B. C1, B. HLJ1, M. CY1, M. SY7, and M. HLJ3 showed high inhibitory activity against the growth of *V. parahaemolyticus*, the inhibition rates of the five strains against *V. parahaemolyticus* were 100%, 100%, 100%, 100%, and 95.04%, respectively. (Figure 1). Similar to the studies reported, the antagonistic activity of the candidate strains against *V. parahaemolyticus* in planktonic and attached forms might be different [22].

Based on the above experimental results, strains B. C1, B. HLJ1, M. CY1, M. SY7, and M. HLJ3 with high inhibitory activity against *V. parahaemolyticus* were selected for the following experiments.

### 3.2. Identification of Strains

The 16S rDNA sequences of B. C1, B. HLJ1, M. C1, M. SY7, and M. HLJ3 were compared by BLAST search in NCBI, and a phylogenetic tree was constructed by the NJ method in the MEGA-X software package. The results showed that M. C1 and *Lactococcus garvieae* (NR104722.1) clustered into the same branch, and M. SY7 clustered with *L. garvieae* (NR113268.1), so the M. C1 and M. SY7 were identified as *L. garvieae*. The M. HLJ3 and *Enterococcus faecalis* (NR115765.1) clustered into the same branch and M. HLJ3 could be identified as *E. faecalis*. Both *L. garvieae* and *E. faecalis* are conditional pathogenic bacteria that cannot be used as aquaculture probiotics. B. C1 and B. HLJ1 could be identified as *Bacillus* by 16S rRNA, and the phylogenetic tree constructed based on the *gyrB* gene could reveal that B. C1 clustered with *B. pumilus* GBSW2 (GU568234.1) (Figure 2A) and B. HLJ1 clustered with *B. subtilis* Y98b (JX513934.1) (Figure 2B). The phylogenetic tree constructed based on the *pycA* gene showed similar results, with B. C1 clustering with *B. pumilus* 3–19 (CP054310) (Figure 2C) and B. HLJ1 clustering with *B. subtilis* NRS6167 (OX419562.1) (Figure 2D). Therefore, B. C1 was identified as *B. pumilus* and B. HLJ1 was identified as *B. subtilis*. B. HLJ1 and B. C1 were selected as candidate probiotics for the following experiments. The partial 16S rDNA sequences of B. C1 and B. HLJ1 were submitted to GenBank, and the accession numbers were assigned as OQ518912 and OQ519649.

### 3.3. Safety of the Candidate Strains

The safety of probiotics is very important for aquaculture; therefore, safety tests including antibiotic resistance and virulence genes need to be conducted on candidate probiotics before application. Antibiotic resistance genes can be transferred between bacteria, leading to the development of antibiotic-resistant bacteria [33]. Thus, antibiotic resistance of probiotic bacteria is a key factor for application in aquaculture. Bernhard et al. [35] demonstrated the transfer of the tetracycline resistance gene from *B. cereus* to *B. subtilis*, the plasmid carrying resistance to tetracycline, pBC16, which was originally isolated from *B. cereus*, could be subsequently transformed in *B. subtilis* and stably maintained. The minimum inhibitory concentration (MIC) of antibiotics against B. C1 and B. HLJ1 were measured using the broth microdilution method, and the susceptibility of the strains to the corresponding antibiotics was assessed using the CLSI guidelines. B. C1 and B. HLJ1 were sensitive to most of antibiotics required by EFSA (FEEDAP 2012), but B. C1 showed intermediate sensitivity to erythromycin and B. HLJ1 showed intermediate sensitivity to chloramphenicol (Table 2). The experimental results indicate that B. C1 and B. HLJ1 have fewer antibiotic resistance factors.

Virulence genes in probiotics is critical because expression of virulence genes can lead to disease outbreaks in aquaculture animals and virulence genes possibly transferred from probiotics to other bacteria [30]. In this study, enterotoxin genes *nheA*, *nheB*, *nheC*, *hblA*, *hblC*, *hblD*, and *entFM* were amplified using PCR. The PCR results were displayed in Table 3. It showed that both B. HLJ1 and B. C1 do not have the enterotoxin genes *nheA*, *nheB*, *nheC*, *hblA*, *hblC*, *hblD*, and *entFM*, indicating that they are safe for application in aquaculture.

### 3.4. Acid Tolerance and Spore Production of the Candidate Strains

The gastrointestinal tract of aquatic animals is an acid environment, and acid tolerance is also one of the evaluation criteria for probiotics in order to survive and colonize the gastrointestinal tract. The acid tolerance assay revealed that B. C1 and B. HLJ1 were highly tolerant to acid environments (Figure 3A). When B. C1 and B. HLJ1 were treated in PBS from pH 4 to 6 for 2 h, the survival rate reached above 96%. Even after treatment at pH 3 for 2 h, the survival rates of B. C1 and B. HLJ1 exceeded 94% and 92%, respectively. It was consistent with previous study, as *Bacillus* sp. YB1701 had a survival rate of 89.9% after treatment for 2 h when the environmental pH was 3 [32].

In aquaculture, probiotics are often added to feeds for use, and high temperatures are often used in feed production to improve palatability and kill pathogens. To prevent inactivation by high temperatures, probiotics must be heat resistant [36]. The tolerance of *Bacillus* to high temperature is attributed to the formation of endospores, which can resist high temperatures and other adverse environments [7]. With longer incubation time, the spore production rates of B. C1 and B. HLJ1 gradually increased. When the incubation time reached 36 h, the spore production rates of B. C1 and B. HLJ1 reached 96% and 92%, respectively (Figure 3B). The spore production rates of B. C1 and B. HLJ1 were significantly higher than those of *B. methylotrophicus* (spore production rate 55%) and *B. subtilis* (spore production rate 80%) reported in the previous study [37,38].

### 3.5. Extracellular Enzyme Activity and Aggregation Capacity

Extracellular enzyme production is an important feature of good probiotics, which provide beneficial effects to the host in terms of food absorption and improve aquaculture water quality by degrading organic waste, including amylase, protease, and cellulase [39,40]. As seen from Table 4, B. HLJ1 was able to produce three extracellular enzymes, including amylase, protease, and cellulase, and B. C1 could produce protease and cellulase. It has been reported that *B. subtilis* SK07 and *B. amyloliquefaciens* SK27 can produce amylase and protease as well as cellulase, and *B. licheniformis* ABSK55 can produce protease and cellulase. When they were added to water or feed at a concentration of 10^9^ CFU/mL, they could promote the growth of shrimp [41].

The auto-aggregation capacity shows a significant correlation with cell adhesions, contributing to the colonization of probiotics in aquaculture animals, and the capacity to co-aggregate with pathogens can help prevent pathogens from invading aquaculture animals [42,43]. After the first 2 h, the auto-aggregation rates of B. HLJ1 and B. C1 were 63.9% and 45.2%, while after 24 h they were 89.8% and 75.3%, respectively (Figure 4A). The auto-aggregation rates of B. HLJ1 and B. C1 were significantly higher than those previously reported for *B. tequilensis*, *B. velezensis*, and *B. subtilis* after the first 2 h (all below 40%), while also having the same efficacy after 24 h [44]. The results indicated that B. HLJ1 and B. C1 had good cell adhesion abilities. The co-aggregation rates of B. HLJ1 and B. C1 with *V. parahaemolyticus* were 43.4% and 36.7% after the first 2 h, while they were 65.6% and 74.3% at 24 h, respectively (Figure 4B). The results were comparable to those previously reported, such as the co-aggregation rate of *B. subtilis* with *Aeromonas hydrophila* being 80% after 24 h [33], indicating the great potential of B. HLJ1 and B. C1 to control *V. parahaemolyticus* infection in aquaculture animals.

### 3.6. Identification of Anti-V. parahaemolyticus Substance

*Bacillus* produces a diverse array of over 20 different types of antimicrobial compounds, including antibiotics, bacteriocins, and lipopeptides [45]. Gao et al. [46] reported that *B. subtilis* H2 produces an anti-*Vibrio* substance that inhibits the growth of 29 *Vibrio* strains by disrupting cell membranes and causing cell lysis, structurally identical to amicoumacin A. The HPLC diagram showed the active peaks where the anti-*V. parahaemolyticus* substances were located (Figure 5A), and the UV (MeOH) spectrum of the anti-*V. parahaemolyticus* compound displayed bands at 315 nm, 247 nm, and 209 nm. The molecular mass of anti-*V. parahaemolyticus* compound produced by B. HLJ1 and B. C1 were analyzed using an LC/MS system, and the (M+H)^+^ ion at *m*/*z* 390.1542 Da and 390.1541 Da (Figure 5B). Comparison with previous studies [47,48,49,50,51], B. HLJ1 and B. C1 were found to produce the same anti-*V. parahaemolyticus* substance, identified as AI-77-F (Figure 5C), which was first isolated and identified from *B. subtilis* AI-77 and belongs to the amicoumacins [52].

### 3.7. Inhibitory Activity against Different V. parahaemolyticus

Aquaculture conditions for different products such as shrimp, fish, crab, and oyster have large or great diversity, and contain different *V. parahaemolyticus*. To effectively control the growth of *V. parahaemolyticus* in different aquaculture environments, it is necessary to test the inhibitory effect of B. HLJ1 and B. C1 on different sources of *V. parahaemolyticus*. Inhibitory activities of B. HLJ1 and B. C1 against 11 different *V. parahaemolyticus* were tested using a double-layer plate method. B. HLJ1 and B. C1 showed inhibitory activity against all *V. parahaemolyticus*, although there were variations in their inhibitory effects against different *V. parahaemolyticus* (Table 5). In exception to *V. parahaemolyticus* ATCC17802 (Figure 6), B. C1 showed high inhibitory activity against *V. parahaemolyticus* from cuttlefish, mantis shrimp, and scallop and showed moderate inhibitory activity against *V. parahaemolyticus* from oyster, forktail, hairtail, grass shrimp, haliotis, and crab. B. C1 also showed low inhibitory activity against *V. parahaemolyticus* from lobster. B. HLJ1 showed moderate inhibitory activity against *V. parahaemolyticus* from cuttlefish and mantis shrimp and showed low inhibitory activity against *V. parahaemolyticus* from oyster, forktail, hairtail, grass shrimp, lobster, haliotis, scallop, and crab. The results indicate the potential of B. HLJ1 and B. C1 to control different sources of *V. parahaemolyticus*.

### 3.8. Anti-V. parahaemolyticus Effect in Simulated Aquaculture Wastewater

In aquaculture, the pathogenic bacteria could cause a disease outbreak when they reach a concentration of approximately 10^4^ CFU/mL [53]. In this study, B. HLJ1 and B. C1 were co-cultured with *V. parahaemolyticus* (10^4^ CFU/mL) at three initial concentrations in simulated aquaculture wastewater, respectively. When the initial concentration of B. HLJ1 was 1 × 10^5^ CFU/mL, it showed low inhibitory activity against the growth of *V. parahaemolyticus*. When the initial concentration of B. HLJ1 was 1 × 10^6^ CFU/mL, it could reduce *V. parahaemolyticus* by more than two orders of magnitude (2.35 lg) at 48 h. When the initial concentration of B. HLJ1 was 1 × 10^7^ CFU/mL, B. HLJ1 reduced *V. parahaemolyticus* by about three orders of magnitude (2.67 lg) at 12 h, by more than four orders of magnitude (4.45 lg) at 24 h, and completely inhibited *V. parahaemolyticus* at 48 h (Figure 7A). However, when the initial concentrations of B. C1 were 1 × 10^5^ CFU/mL and 1 × 10^6^ CFU/mL, it showed low inhibitory activity against the growth of *V. parahaemolyticus* at 12 h and 24 h, and completely inhibited *V. parahaemolyticus* at 48 h. When the initial concentration of B. C1 was 1 × 10^7^ CFU/mL, B. C1 reduced *V. parahaemolyticus* by about two orders of magnitude (1.99 lg) at 12 h, by more than four orders of magnitude (4.14 lg) at 24 h, and completely inhibited *V. parahaemolyticus* at 48 h (Figure 7B). The inhibitory activity against *V. parahaemolyticus* was influenced by the concentration, and this finding supports previous research that the number of antagonistic bacteria must be higher than the number of pathogens for effective control [54].

## 4. Discussion

The long-term use of antibiotics has caused aquaculture to face numerous challenges, including the development of antibiotic-resistant bacteria, drug residues, and environmental contamination [6]. Therefore, finding alternative methods of disease prevention is essential for sustainable aquaculture [55]. According to the research, there is a promising prospect of replacing antibiotics with probiotics in aquaculture [56].

In this study, two probiotics that inhibit *V. parahaemolyticus* were isolated from aquaculture water (B. C1 and B. HLJ1). B. C1 and B. HLJ1 were identified as *B. pumilus* and *B. subtilis* by using the *gyrB* and *pycA* sequences. Many studies have reported the inhibitory effect of *Bacillus* on pathogenic bacteria [7]. The probiotic *B. licheniformis* exhibits strong inhibitory activity against *A. hydrophila* and feeding diets supplemented with *B. licheniformis* (10^8^ CFU/g of feed) improved the immunity of *Labeo rohita* and its resistance to *A. hydrophila* [57]. Another study found that *B. velezensis* had inhibitory effects on *V. parahaemolyticus*, *A. veronii*, and *Plesiomonas*; the inhibition effect increases with the rising concentration of *B. velezensis* and when the concentration reaches 2.7 × 10^8^ CFU/mL, the inhibition effect is significant [58]. In this study, B. C1 and B. HLJ1 effectively inhibited *V. parahaemolyticus* at a concentration of 10^7^ CFU/mL. The inhibition activity against *V. parahaemolyticus* was influenced by the concentration, a finding similar to that previously reported [54].

In aquaculture, probiotics should not carry antibiotic-resistant genes and be non-pathogenic to aquaculture animals [59]. It has been demonstrated that antibiotic-resistant bacteria transfer resistance genes to other bacteria, resulting in the emergence of more and more antibiotic-resistant bacteria [33]. In this study, both B. C1 and B. HLJ1 were sensitive to most of antibiotics recommended by EFSA (FEEDAP 2012). Probiotics cannot contain virulence genes because bacteria carrying virulence genes may cause disease outbreaks in aquaculture animals [30]. The results of the virulence gene detection experiment revealed that B. C1 and B. HLJ1 do not have the enterotoxin genes (*nheA*, *nheB*, *nheC*, *hblA*, *hblC*, *hblD*, and *entFM*). The safety shown by B. HLJ1 and B. C1 makes them prospective for use as probiotics in aquaculture.

Heating is frequently used in aquaculture feed production to improve palatability and kill pathogens [36]. Therefore, probiotics must be heat-resistant. The heat resistance of *Bacillus* is mainly due to its ability to produce endospores, so the heat resistance of *Bacillus* is generally assessed by evaluating its ability to produce spores [7]. In this experiment, both B. C1 and B. HLJ1 had high spore production rates. When incubated in LB medium for 36 h, the spore production rates of B. C1 and B. HLJ1 reached 96% and 92%, respectively. Probiotics should be acid-tolerant in order to colonize the gastrointestinal tract of aquatic animals. Even after 2 h of treatment at pH 3, the survival rates of B. C1 and B. HLJ1 exceeded 94% and 92%, respectively. The results of the experiments revealed that B. C1 and B. HLJ1 have a high tolerance to acid environments.

The production of extracellular enzymes is one of the criteria for assessing probiotics, which promote the growth and development of aquatic animals as well as improve the quality of aquaculture water [39,40]. It has been reported that *B. methylotrophicus* can produce amylase, lipase, and protease, promoting the growth of fish [60]. In this study, B. HLJ1 can produce protease, cellulase, and amylase, and B. C1 can produce protease and cellulase. The results suggest that both strains have probiotic potential. Canzi et al. [42] found that co-aggregation of probiotics and pathogens could reduce pathogens attachment in the digestive tract of aquaculture animals. In addition, there was a strong correlation between the auto-aggregation ability of probiotics and their adhesion to aquaculture animals [43]. In this study, B. C1 and B. HLJ1 showed good auto-aggregation capacity and co-aggregation capacity with *V. parahaemolyticus*. This finding is consistent with a previous study showing that *Bacillus* has the ability to auto-aggregate and co-aggregate with *A. hydrophila* and *E. faecalis* [61].

One of the main mechanisms by which probiotics inhibit pathogens is through the secretion of antibacterial substances [62]. Based on previous studies [47,48,49,50,51] and MS analysis, the anti-*V. parahaemolyticus* substances produced by B. C1 and B. HLJ1 were identical to AI-77-F. The AI-77 series compounds, first isolated and identified from *B. subtilis* AI-77 by Shimojima et al. [52], belong to the amicoumacins. Huang et al. [49] isolated AI-77-F from *Alternaria tenuis* Sg17-1 and investigated its inhibitory effect on human malignant tumor A375-S2 and Hela cells, but the effect was very weak. Shi et al. [51] isolated AI-77-F from *B. subtilis* subsp. Inaquosorum and found that it had potential inhibitory activity against quorum sensing (QS) and could be used in the development of a novel anti-microbial agent to treat pathogenic infections. In this study, it is the first report that AI-77-F has inhibitory activity against *V. parahaemolyticus*.

Compared to laboratory conditions, aquaculture environments are more complex, nutrient-poor, and contain different *V. parahaemolyticus*. Both B. C1 and B. HLJ1 showed inhibitory activity against 11 different *V. parahaemolyticus* and could effectively control the growth of *V. parahaemolyticus* in simulated aquaculture wastewater when the concentration reached 1 × 10^7^ CFU/mL. The results showed their great potential as probiotics for application in different aquaculture environments to control *V. parahaemolyticus*.

In conclusion, B. HLJ1 and B. C1 have inhibitory activity against *V. parahaemolyticus*, and the most effective dose of both probiotics B. C1 and B. HLJ1 to control *V. parahaemolyticus* was 10^7^ CFU/mL. Based on the most effective doses of B. C1 and B. HLJ1 against *V. parahaemolyticus* and the commercial market price in China, the cost of using B. C1 and B. HLJ1 for effective control of *V. parahaemolyticus* in aquaculture can be estimated to be about USD 0.0241/m^3^ [63]. The safety tests indicated that they were safe for application in aquaculture. B. HLJ1 and B. C1 can also tolerate acid environments, produce spores, secrete extracellular enzymes, and co-aggregate as well as auto-aggregate with *V. parahaemolyticus*. As a result, there is great potential for using B. HLJ1 and B. C1 as aquaculture probiotics.

## Figures and Tables

**Figure 1 microorganisms-11-01667-f001:**
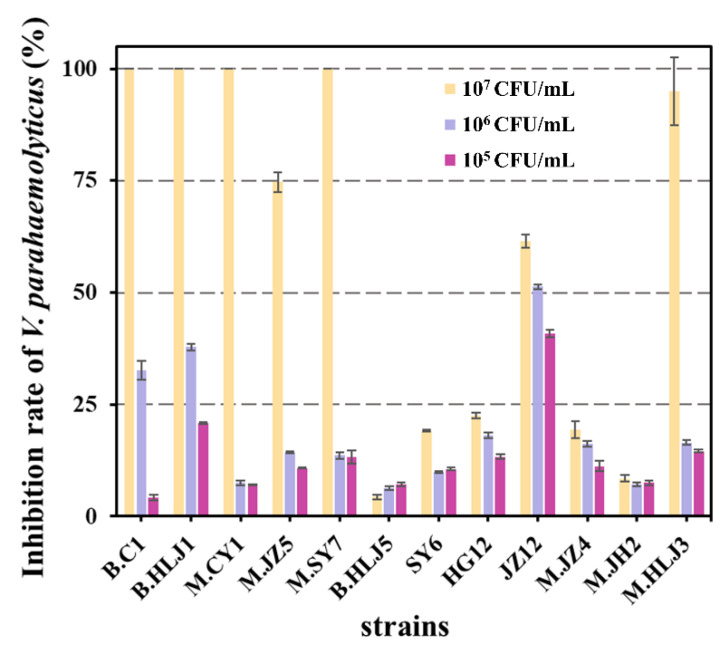
Assessment of inhibitory effect of isolated strains on *V. parahaemolyticus*. The concentration of *V. parahaemolyticus* was 1 × 10^4^ CFU/mL and the concentrations of isolated strains were 1 × 10^7^ CFU/mL (Yellow), 1 × 10^6^ CFU/mL (Purple), and 1 × 10^5^ CFU/mL (Pink), respectively.

**Figure 2 microorganisms-11-01667-f002:**
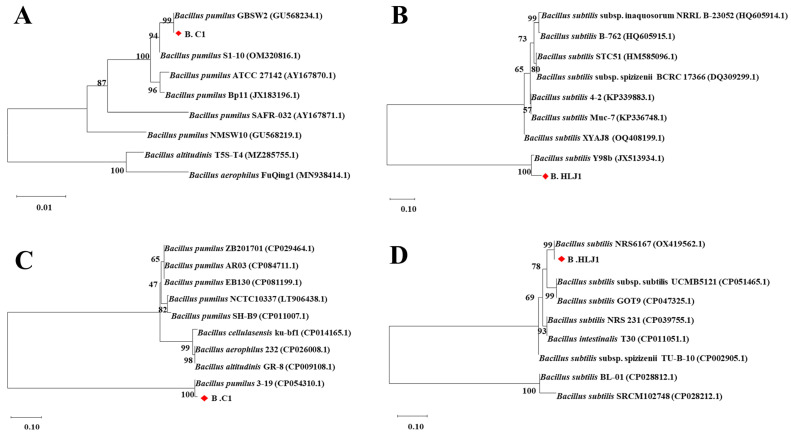
Phylogenetic tree of B. C1 and B. HLJ1 using NJ method. (**A**) B. C1 based on *gyrB* gene, (**B**) B. HLJ1 based on *gyrB* gene, (**C**) B. C1 based on *pycA* gene, and (**D**) B. HLJ1 based on *pycA* gene.

**Figure 3 microorganisms-11-01667-f003:**
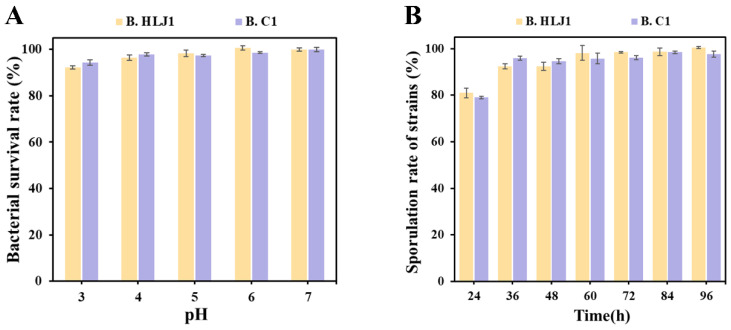
Acid tolerance (**A**) and spore production (**B**) of B. HLJ1 and B. C1.

**Figure 4 microorganisms-11-01667-f004:**
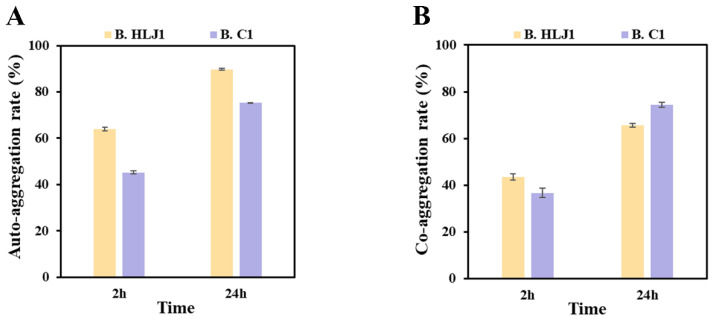
Aggregation capacity of B. HLJ1 and B. C1. (**A**) Auto-aggregation rate, (**B**) co-aggregation rate with *V. parahaemolyticus*.

**Figure 5 microorganisms-11-01667-f005:**
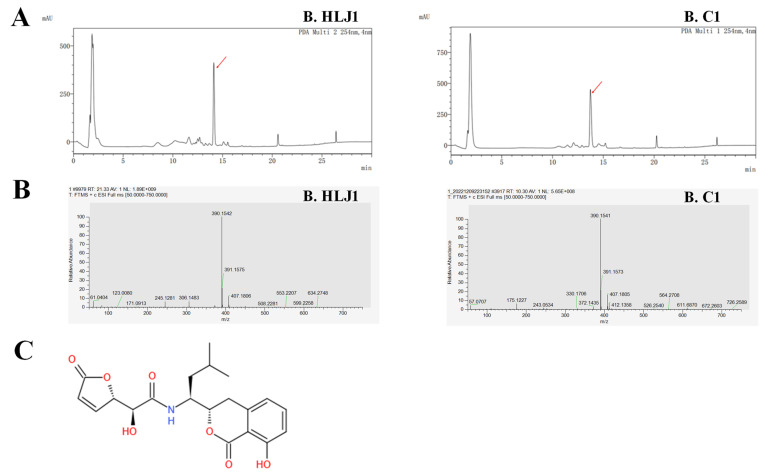
Identification of anti-*V. parahaemolyticus* substance secreted by two potential probiotics. (**A**) HPLC diagram of the anti-*V. parahaemolyticus* compound from B. HLJ1 and B. C1 (the red arrow indicates the peak where the active compound is located), (**B**) MS analysis of anti-*V. parahaemolyticus* compound from B. HLJ1 and B. C1, (**C**) chemical structure of anti-*V. parahaemolyticus* compound, which was identified as AI-77-F.

**Figure 6 microorganisms-11-01667-f006:**
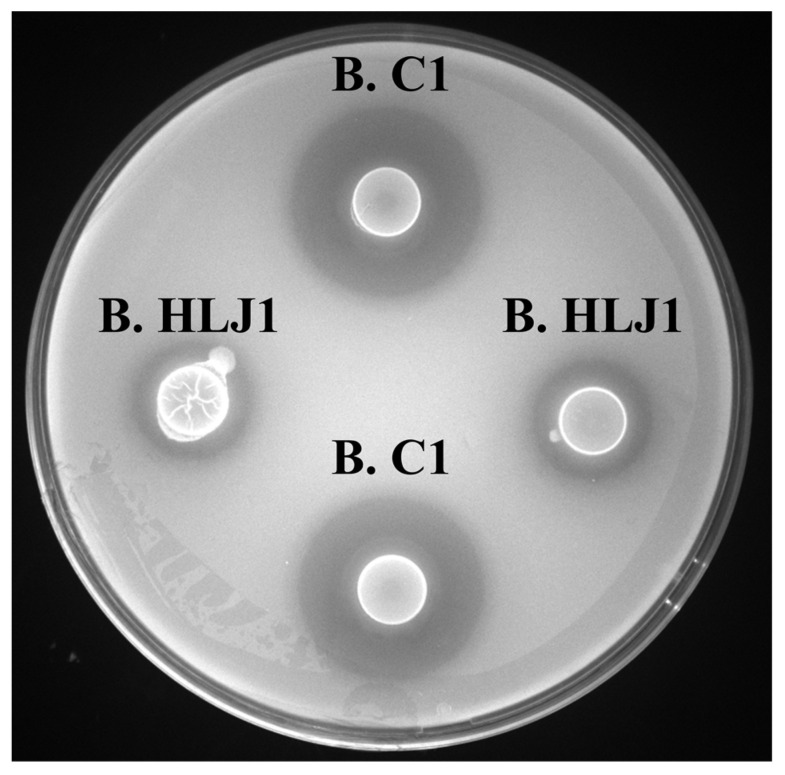
Inhibition zone (mm) by B. C1 and B. HLJ1 against *V. parahaemolyticus* 17802.

**Figure 7 microorganisms-11-01667-f007:**
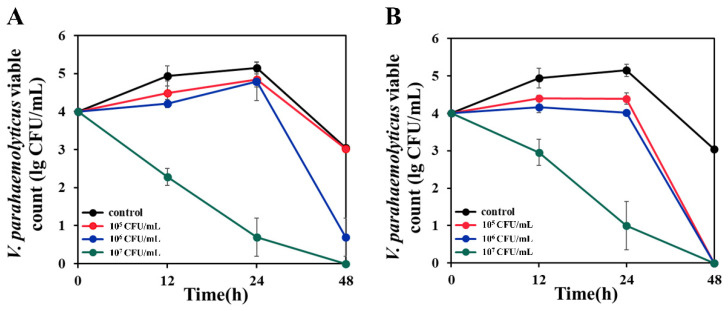
Anti-*V. parahaemolyticus* effect of two strains in simulated aquaculture wastewater. (**A**) B. HLJ1, (**B**) B. C1. The initial concentrations of B. HLJ1 and B. C1 were 10^5^ CFU/mL (Red), 10^6^ CFU/mL (Blue), 10^7^ CFU/mL (Green).

**Table 1 microorganisms-11-01667-t001:** Inhibitory activity of selected bacteria against *V. parahaemolyticus*.

Strains	Inhibitory Activity	Strain	Inhibitory Activity
B. C1	++++	M. CY1	+++
B. HLJ1	+++	M. JZ4	+++
B. HLJ5	++	M. JZ5	++
SY6	++	M. JH2	++
HG12	++	M. SY7	++
JZ12	++	M. HLJ3	++

Note: ++++ represents the diameter of the inhibition circle >20 mm, +++ represents the diameter of the inhibition circle >15 mm and ≤20 mm, ++ represents the diameter of the inhibition circle >10 mm and ≤15 mm.

**Table 2 microorganisms-11-01667-t002:** Antibiotic resistance of B. C1 and B. HLJ1.

Antibiotic	Judgement Standard (μg/mL)			B. C1		B. HLJ1	
	S	I	R	MIC (μg/mL)	Sensitivity	MIC (μg/mL)	Sensitivity
Teicoplanin	8	16	32	0.25	S	0.25	S
Vancomycin	4	8–16	32	0.25	S	0.25	S
Chloramphenicol	8	16	32	8	S	16	I
Tetracycline	4	8	16	0.25	S	0.25	S
Erythromycin	0.5	1–4	8	1	I	0.25	S
Gentamycin	4	8	16	4	S	2	S
Kanamycin	16	32	64	16	S	2	S

Note: According to the guidelines of the CLSI, strains were classified as sensitive (S), intermediate (I), or resistant (R).

**Table 3 microorganisms-11-01667-t003:** Distribution of virulence genes in B. C1 and B. HLJ1.

Strain	Virulence Genes						
	*hblA*	*hblC*	*hblD*	*nheA*	*nheB*	*nheC*	*entFM*
B. C1	−	−	−	−	−	−	−
B. HLJ1	−	−	−	−	−	−	−

Note: (−) represents the absence of a gene.

**Table 4 microorganisms-11-01667-t004:** Extracellular enzyme production abilities of B. HLJ1 and B. C1.

Strain	Ratio of Enzymatic Circle Diameter to Colony Diameter		
	Amylase	Protease	Cellulase
B. HLJ1	1.33 ± 0.13	1.27 ± 0.11	1.23 ± 0.06
B. C1	−	2.08 ± 0.32	2.14 ± 0.26

**Table 5 microorganisms-11-01667-t005:** Inhibitory activity against different *V. parahaemolyticus*.

Strains	Origin	Diameter of Inhibition Zone (mm)	
		B. C1	B. HLJ1
*V. parahaemolyticus* ATCC17802	American Type Culture Collection	23.3 ± 0.3	15.3 ± 0.3
*V. parahaemolyticus* OY14	American oyster	16.5 ± 0.5	11.5 ± 0.5
*V. parahaemolyticus* FFTF11	Frozen fork tail fillets	15.5 ± 0.5	12.0 ± 0.1
*V. parahaemolyticus* BC21	Bengali frozen cuttlefish	22.0 ± 1.0	19.5 ± 0.5
*V. parahaemolyticus* IFH23	Indonesian frozen hairtail	17.5 ± 0.5	11.5 ± 0.5
*V. parahaemolyticus* TSG36	Thai grass shrimp	18.5 ± 1.0	11.5 ± 0.5
*V. parahaemolyticus* SAL38	South African lobster	11.4 ± 0.1	10.3 ± 0.3
*V. parahaemolyticus* HA44	Haliotis	19.3 ± 0.2	11.1 ± 0.1
*V. parahaemolyticus* TMS61	Thai mantis shrimp	20.1 ± 0.2	15.2 ± 0.1
*V. parahaemolyticus* SC123	Scallop	21.0 ± 0.5	14.5 ± 0.5
*V. parahaemolyticus* CR127	Crab	19.5 ± 0.5	14.0 ± 0.4

Note: Values are presented as mean ± SD. The diameter of the inhibition circle >20 mm represents the high inhibitory activity, the diameter of the inhibition circle >15 mm and ≤20 mm represents the moderate inhibitory activity, the diameter of the inhibition circle >10 mm and ≤15 mm represents the low inhibitory activity.

## Data Availability

The raw data supporting the conclusion of this article will be made available by the authors, without undue reservation.

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
