# Peer review of "Isolation of *Bacillus subtilis* and *Bacillus pumilus* with Anti-*Vibrio parahaemolyticus* Activity and Identification of the Anti-*Vibrio parahaemolyticus* Substance"

_microorganisms, 2023, doi:10.3390/microorganisms11071667_

Round 1

Reviewer 1 Report

This manuscript describes Bacillus strains isolated from aquaculture water, that showing antimicrobial activity to Vibrio parahaemolyticus.

This includes lots of data, and data can be useful for aquaculture application, but this requires some improvement for the publication.

in 2.2.1. Primary screening, please specify the Vibrio strain you used. Give more information. It this Vibrio virulent in aquaculture? and carry any virulence gene? If not, it is useless to inhibit non virulent Vibrio.

in 2.3. author used 16S rRNA sequencing, but this it not a ideal method for Bacillus identification. Please examine additional experiments for identification.

in section 2.4. Why did you use only 2 strain? Even B. HLJ1 Bacillus is not showing the strongest antimicrobial activity.

2.5.2. for spore production assay, I am not sure that 37C is good for sporulation. 

Please describe relationship between Table 1 and Figure 1.

No comments

Reviewer 2 Report

The manuscript is well written and the data are well- presented. However, some points need to be improved such as:

1- The introduction should be supported with a paragraph about the economic losses associated with V. parahaemolyticus infection in aquaculture.

2-  All the abbreviations should  firstly mentioned in the full names, like MRS agar, MIC, CLSI, etc.

3- The result of sensitivity test  of -V. parahaemolyticus to different isolate should be supported with photos, showing  the inhibition zones.

4- The cost of using these probiotics commercial practices and also, their most effective dose  should be mentioned in rhe conclusion section  

The quality of English language is good, just minor check of the grammer of some sentences should be done.

Round 2

Reviewer 1 Report

No additional comment

No additional comment

Author Response

Thanks again for your review and good luck!